# Evaluation of the immune status of dogs vaccinated against rabies by an enzyme-linked immunosorbent assay using crude preparations of insect cells infected with a recombinant baculovirus encoding the rabies virus glycoprotein gene

A. K. Santosh[1], Deepak Kumar[2], Charanpreet Kaur[3], Priya Gupta[3,4], Pagala Jasmeen[3,4], L. Dilip[1], G. Kavitha[1], Suresh Basagoudanavar[5], Madhusudan Hosamani[5], V. Balamurugan[6], R. Sharada[1], D. Rathnamma[1], K. M. Sunil[1], Nagendra R. Hegde[3,4]*, Shrikrishna Isloor[1]*

1 KVAFSU-CVA Rabies Diagnostic Laboratory, Department of Veterinary Microbiology, Veterinary College, Karnataka Veterinary Animal and Fisheries Sciences University, Bengaluru, India, 2 Ella Foundation, Genome Valley, Turkapally, Shameerpet Mandal, Hyderabad, India, 3 National Institute of Animal Biotechnology, Gachibowli, Hyderabad, India, 4 Regional Centre for Biotechnology, Faridabad, India, 5 ICAR – Indian Veterinary Research Institute, Bengaluru, India, 6 ICAR – National Institute of Veterinary Epidemiology and Disease Informatics, Bengaluru, India

* hegde@niab.org.in (NRH); kisloor@gmail.com (SI)

## Abstract

Evaluation of the effectiveness of vaccination of animals against rabies is not routinely implemented. In cases where it is carried out, the rapid fluorescent focus inhibition test (RFFIT) or the fluorescent antibody virus neutralization (FAVN) test are the recommended tests. However, both of these tests require handling of live rabies virus (RABV), and are cumbersome to perform. In view of this, the enzyme-linked immunosorbent assay (ELISA) has been proposed as a surrogate test; however, availability of appropriate antigen is a major impediment for the development of ELISAs to detect anti-rabies antibodies. The most widely used antigen is the RABV glycoprotein (G) purified from cell culture-propagated virus, which requires a biosafety level 3 containment. The alternative is to use recombinantly expressed G, which needs to be to be properly glycosylated and folded to serve as the best antigen. The most suitable system for its production is the baculovirus expression system (BVES). However, purification of RABV G is challenging. We therefore tested partially purified preparations in the form of extracts of insect cells infected with baculovirus expressing RABV G, against sera from vaccinated dogs in an indirect ELISA. The results showed good concordance against RFFIT, with sensitivity and specificity of 90.48% and 80.00%, respectively. The system may be used for quick screening to determine the presence and an approximate level of antibodies, and can be modified to enable monitoring of mass dog vaccination programs, as well as to facilitate certification of dogs intended for international travel and transportation.

**Data Availability Statement:** All relevant data are within the manuscript and its Supporting information files.

**Funding:** This research was funded in part by the Department of Biotechnology, Ministry of Science and Technology, Government of India (BT/ADV/Canine Health/GADVASU/2017-18). The funders had no role in the design of the study; in the collection, analyses, or interpretation of data; in the writing of the manuscript, or in the decision to publish the results.

**Competing interests:** The authors have declared that no competing interests exist.

# Introduction

Rabies has existed for at least 4000 years, and continues to pose a serious public health threat, particularly for children aged <15 years, especially in Asia and Africa [1, 2]. This nearly 100% fatal disease, caused by *Lyssavirus rabies* (RABV), is completely preventable through vaccination, even after exposure [3–5]. Dogs account for 54%, terrestrial wildlife (skunks, raccoons, foxes, cats, etc.) for 42%, and bats 4% of the human rabies cases worldwide [6, 7]. However, in many low- and middle-income countries (LMIC) where canine rabies has not been adequately controlled, dogs account for at least 90% of the human rabies cases [1, 8]. Consequently, vaccinating domestic dogs is the most economical and impactful intervention in preventing rabies in humans [5, 7, 9]. It has been estimated that vaccinating >70% of the dogs would be sufficient to create a barrier which would block transmission of RABV to humans [10, 11]. However, whilst recording vaccination is possible for pets, the same is difficult to impossible for free ranging dogs, especially in LMIC, and hence there is no means to verify the success of the vaccination programmes other than keeping a record of the vaccination. Therefore, knowing the immune status by estimating the level of antibodies in vaccinated dogs would be vital, both in the case of vaccination drives, particularly in urban areas, as well as for certification of dogs for international transportation. In addition, monitoring the level of antibodies would be required when clinical trials are conducted for new vaccines or when new vaccination regimen are being standardized.

Presence of neutralising antibodies (NAbs) is the most commonly studied correlate of protection, and hence the most reliable indicator of the success of vaccination against several virus diseases. Detection and quantification of NAbs is typically achieved by employing the virus neutralization (VN) test. For a number of years, the mouse neutralization test (MNT) was the only test used for assessing the presence as well as quantification of NAbs against RABV [12, 13]. Owing to the ethical issues of using mice, the associated expenditure and the time involved, cell culture techniques were later developed. The most widely used techniques are those based on the reduction in the proportion of RABV-infected cells by fluorescence methods, namely, the rapid fluorescent focus inhibition test (RFFIT) and the fluorescent antibody virus neutralization (FAVN) test [14, 15]. These tests are typically used for surveillance in animals (dog, fox) either before or after vaccination. An antibody titre of at least 0.5 international units per mL (IU/mL) of serum is considered to be adequate to confer protection to dogs and other carnivores [4, 16–19], although some dogs show protection without NAbs [20].

In spite of their immense utility, both RFFIT and FAVN are time-consuming, expensive, not amenable to high-throughput testing, and require a fluorescent microscope, trained personnel as well as high-level bio-containment (to handle live RABV), not to mention that they are difficult to standardize across laboratories. The read-out also involves observation of at least 20 fields or calculating the proportion of infected cells for each dilution of the antibody, both of which may result in objectivity and varying reproducibility. In addition, poor quality sera (e.g., haemolysed) can cause toxicity to cells used in the assay, making the interpretation of the results difficult [21, 22]. For these reasons, various other techniques have been explored as alternatives. Neutralization of pseudotyped lentiviruses has been shown to be as good as RFFIT [23]. However, the limitations of RFFIT and FAVN, except for the requirement for higher level biocontainment, apply to this test as well. A simpler, user-friendly test is the enzyme immunoassay, a variety of which have been described to quantify the level of antibodies in humans, domestic animals and wildlife species [22, 24–48]. Among these, the enzyme-linked immunosorbent assay (ELISA) is easy and rapid (~4 h) to perform, and does not require highly trained workers, nor handling of live virus, and can be performed even with poor quality sera [26, 29, 30, 49]. ELISA has now been accepted for use in the verification of the immune

status of dogs and cats as a pre-requisite for international movement, provided that the test is validated as fit for such purposes.

Commercial ELISA kits have been developed for detecting antibodies in dogs and other animal species. However, such kits are cost prohibitive, limiting their routine usage, especially in resource-limited settings. Moreover, most such kits use the RABV glycoprotein (G) purified from bulk propagated infectious virus, a process which is time-consuming and requires bio-safety level 3 facilities. These limitations could be overcome by using recombinant G as the antigen. The RABV G has been expressed using various systems, both for use as a vaccine candidate as well as for use as an antigen in diagnostic assays. Full-length and truncated G have been expressed in bacteria and yeast. However, bacterially expressed G is often misfolded, toxic to bacteria, unglycosylated, and poorly immunogenic, and that expressed in yeast appears to undergo improper glycosylation as well as abnormal folding and oligomerization. On the other hand, RABV G expressed in eukaryotic cells of plant or animal origin has been reported to be properly folded and glycosylated (reviewed in [50]).

The best substrates for producing recombinant G are mammalian cells, but they require expensive medium and supplements for their propagation, are mostly anchorage-dependent, do not attain a high density, and are sensitive to hydrodynamic stress [51], making large-scale antigen production through the use of mammalian cells cumbersome and expensive. On the other hand, production of antigen in insect cells through baculoviruses is more economical. Insect cells require relatively milder culture conditions, such as propagation at lower temperatures and no requirement for $CO_2$. In addition, the baculoviruses are safe for handling since their tropism is restricted to insects. The cells and viruses can also be propagated in serum-free media and can easily be scaled up. When mammalian or viral proteins are expressed in insect cells, protein folding and processing are more native than when expressed through prokaryotic systems [52, 53]. The RABV G expressed using baculovirus expression system (BVES) has been shown to be similar structurally and biologically to the native protein and has been used as an immunogen [54–57].

Even when a recombinant protein is expressed to reasonable levels, whether in mammalian cells or through BVES, it is often difficult to recover and purify membrane-anchored, glycosylated proteins. Purification of native RABV G from virus propagated in mammalian cells is well-established [58] but requires virus culture. On the other hand, purification of G with the retention of its trimeric, fully glycosylated, and immunogenic structure when expressed in eukaryotic cells is not a trivial task (reviewed in [50]). With the intent to avoid purification processes which are either complicated or result in poor recovery, we used crude preparations of baculovirus-infected insect cells for the development of an indirect ELISA (iELISA) for the detection and semi-quantitative estimation of anti-RABV antibodies in dogs vaccinated against rabies.

## Material and methods

### Ethics statement

As per the rules and guidelines of the Committee for the Purpose of Control and Supervision of Experiments on Animals (CPCSEA; recently renamed CCSEA), Department of Animal Husbandry and Dairying, Ministry of Fisheries, Animal Husbandry and Dairying, Government of India, no ethical clearance is required for the collection of blood samples of volumes up to 10 mL from animals for academic research purposes in India.

### Serum samples

Serum samples (n = 245) of vaccinated pet dogs submitted to the Rabies Diagnostic Laboratory, Veterinary College, Karnataka Veterinary & Animal Fisheries Sciences University,

Bengaluru, India, were used in the current study. The sera had been collected from ten different states of India, namely, Maharashtra (n = 107), Karnataka (n = 87), Delhi (n = 13), Punjab (n = 10), Haryana (n = 9), Andhra Pradesh, Kerala (n = 6 each), Gujarat, Tamil Nadu, and West Bengal (n = 3 each). The serum samples were stored at -20°C until used for testing.

## Cell and virus culture

Baby hamster kidney– 21 (BHK-21; American Type Culture Collection, USA) cells were propagated at 37°C and 5% $CO_2$ in Dulbecco's modified Eagle's medium (DMEM) supplemented with 5% fetal bovine serum (FBS; Gibco, USA), 100 μg/ml of penicillin and 100 U/ml of streptomycin (Invitrogen, USA). The *Spodoptera frugiperda* insect-derived Sf21 cells were propagated at 25°C without $CO_2$ in Sf900 II medium (Thermo Fisher Scientific, USA) supplemented with 10% FBS, 100 ug/ml of penicillin and 100 U/ml of streptomycin.

The Larghi strain of RABV was propagated by infecting 75–80% confluent BHK-21cells with a multiplicity of infection (MOI) of 0.1. After 48 hrs, the infected cell supernatant was harvested. The recombinant baculoviruses (generated as outlined below) were propagated by infecting 75–80% confluent Sf21 cells at an MOI of 0.1, and harvesting the cells after 48 hrs. With both the viruses, titres were determined with the relevant cells cultured in 96-well tissue-culture plates, and calculating 50% tissue-culture infective dose ($TCID_{50}$) by applying the Reed and Muench method [59].

## Rapid fluorescent focus inhibition test (RFFIT)

The RFFIT was performed as per the protocol described previously [60]. Briefly, test sera were heat inactivated (56°C, 30 min) and diluted serially two-fold in a 96-well tissue culture plate, and mixed with equal volume (100 μL) of RABV diluted to 100 $TCID_{50}$. The WHO reference serum was included in each test. The plate was incubated at 37°C for 90 min in a 5% $CO_2$ incubator. Then, 2.5–3.0 x $10^4$ BHK-21 cells were added to each well and incubated further at 37°C under 5% $CO_2$ for 48 hrs. The medium was decanted from the plate and the cells were fixed in 75% ice-cold acetone for 30 min at -20°C. After decanting the acetone and air drying, the fixed monolayer was reacted with fluorescein isothiocyanate (FITC)-conjugated anti-RABV-nucleocapsid protein antibody, and the plate was incubated at 37°C under 5% $CO_2$ for 60 min. The conjugate was decanted, and the plate was washed two times with phosphate buffered saline (PBS, pH 7.0). The plate was observed under a fluorescent microscope. For each serum sample, the highest dilution that neutralized the virus was recorded. The results were compared to the WHO reference serum (2 IU/mL) which at 1:8 dilution completely neutralized RABV. The titre of anti-RABV antibodies in sera was calculated by using the formula:

Neutralizing reciprocal of test serum showing complete antibody titer in serum

$$= \frac{\text{Neutralization of virus infectivity x Unitage of reference serum}}{\text{Reciprocal of highest dilution of reference serum showing complete neutralization of virus infectivity}}$$

## Generation of recombinant baculovirus encoding RABV G

The *RABV G* gene (Larghi's strain), which was available in pET-32 vector, was re-cloned by including a hexa-histidine tag at the C-terminus, using appropriate primers; additional primers were used to confirm the presence of the gene (Table 1). The *G* gene was inserted into the baculovirus shuttle plasmid pFastbacI using *EcoR*I and *Sal*I enzymes and T4 DNA ligase. The ligated product was transformed into *E. coli* DH5α cells and plated on Luria Bertini (LB) agar

**Table 1. Primers used in the present study.**

| Primer | Sequences (5'-3') | Amplicon size (bp) | Reference |
|---|---|---|---|
| **VF (EcoRI)** | GCG GAA TTC ATG GTT CCT CAG GTT CTT TTG T | 1611 bp | [61] |
| **VR (NotI)** | CCCGG GCG GCC GC TCA CAG TCT GGT CTC ACC CCC | | |
| **VR (SalI) RAB His RER** | TATTGAGTCGACTTATCAGTGATGGTGATGGTGATGACTGCC | 1634 | This report |
| **pFastBac^TM For** | GGATTATTCATACCGTCCCA | | [62] |
| **pFastBac^TM Rev** | CAAATGTGGTATGGCTGATT | | [62] |
| **pUC/M13 Forward** | CCCAGTCACGACGTTGTAAAACG | 3914 | [62] |
| **pUC/M13 Reverse** | AGCGGATAACAATTTCACACAGG | 3914 | [62] |

Note: The underlined sequences represent restriction enzyme recognition sites.

with 50 µg/mL ampicillin, and colonies were confirmed by PCR with pFastBacI forward and reverse primers (listed in Table 1). The pFastBacI-RABV-G was transformed into *E. coli* DH10Bac cells and plated on LB agar containing 50 µg/mL of kanamycin, 7 µg/mL of gentamicin, 10 µg/mL of tetracycline, 100 µg/mL of Blue-gal and 40 µg/mL isopropyl-β-thiogalactoside (IPTG). Positive (white) colonies were selected, and the bacmid was amplified and analysed by PCR using M13 forward and reverse primers (listed in Table 1).

The recombinant bacmid was transfected into Sf21 cells using the Cellfectin^TM reagent as per the manufacturer's instructions. The resultant baculovirus (Passage 1 or P1) was stored at 4˚C, and $TCID_{50}$ was determined as described above. Further propagation to generate P2 and P3 was carried out by infecting Sf21 cells at an MOI of 0.1, and the preparations were titrated and stored at -80˚C for further use.

## Reverse transcription polymerase chain reaction (RT-PCR)

Approximately $0.8 \times 10^6$ Sf21 cells were seeded into each well of a 6-well plate and incubated at 28˚C. The following day, the cells were infected with the recombinant baculovirus or a control baculovirus at an MOI of 0.1 and further incubated at 28˚C for 48 hrs. The cells were harvested, and the supernatant and the pellet were processed separately using Trizol (Invitrogen, USA) to extract RNA. The RNA (1 µg) was utilized as a template for cDNA synthesis using the PrimeScript First Strand cDNA synthesis kit (Takara, Japan). The PCR was performed using gene-specific primers, involving denaturation at 98˚C for 30 sec, annealing at 58˚C for 45 sec, and extension at 72˚C for 90 sec, followed by a final extension at 72˚C for 10 min. The amplified products were analyzed by agarose gel electrophoresis, stained with ethidium bromide, and visualized under UV transillumination in a gel documentation system.

## Western blotting

For confirming the expression of the G protein, supernatants and extracts from baculovirus-infected Sf21 cells were subjected to sodium dodecyl sulphate polyacrylamide gel electrophoresis (SDS-PAGE) and western blotting. For this, $2X10^6$ Sf21 cells were seeded in a 60-mm cell culture dish, incubated at 25˚C without $CO_2$. After an overnight culturing, the cells were infected at an MOI of 10 with the baculovirus encoding RABV G. After 48 hrs, the cells were harvested and centrifuged at 1000 X *g*. The cell pellet was solubilised with lysis buffer [25 mM Tris-HCl pH 7.4, 150 mM NaCl, 1 mM ethylene diamine tetraacetic acid (EDTA), 2.0% Triton X-100, 5% glycerol, 1 mM phenyl methyl sulfonyl fluoride (PMSF) and 8 M urea]. Protein was estimated in various samples by bicinchoninic acid (BCA) assay (Pierce, USA), and equal amounts of protein were electrophoresed on a 12% SDS-PAG, followed by transfer

to a polyvinylidene difluoride (PVDF) membrane. The membranes were blocked with 5% bovine serum albumin in PBS, and immunoblotted using a 1:10,000 dilution of anti-His antibody (Sigma Aldrich, USA) or a 1:300 dilution of a dog serum with an estimated titre of 16 IU of anti-RABV antibodies. Then, the blot was incubated with a 1:5000 dilution of horse radish peroxidase (HRP) conjugated anti-mouse or anti-canine IgG (both from Sigma Aldrich, USA). Between each step, the membranes were washed thrice with PBS containing 0.05% Tween-20. Finally, bands were visualized by using the enhanced chemiluminescence reagent (Pierce, USA) and exposure to an X-ray film or capture through a gel documentation system.

## Enzyme-linked immunosorbent assay

A crude preparation of RABV G was prepared by infecting Sf21 cells with an MOI of 10 of the recombinant baculovirus and harvesting at 96 hrs post-infection. The infected cells were pelleted, and extracts were prepared by one cycle of freeze-thaw. The total protein was estimated by Bradford assay (Bio-Rad, USA), before using it for the iELISA. A checker-board titration was performed and an antigen concentration of 500 ng/well, a serum dilution of 1:100 and a dilution of 1:15,000 for anti-dog IgG-HRP conjugate were determined to be optimum for use in the iELISA.

For testing the sera by iELISA, antigen (in the form of crude extract) was coated in carbonate-bicarbonate buffer (pH 9.6) at 4°C overnight. The content of the wells was discarded, and the plates were washed two times with PBS. The test sera were diluted 1:100 and added in duplicate wells. The appropriate controls *viz*., positive control serum (RFFIT titer of 8 IU/mL when undiluted and diluted 1:2 to 1:64 to achieve 4.0 to 0.125 IU/mL, respectively) negative control serum (from an unvaccinated healthy dog), conjugate controls (no sera added) and blank wells were maintained in each plate. The plate was then incubated at 37°C for 2 hrs. The plate was washed with PBS containing 0.05% Tween-20 (PBST) and horse radish peroxidase (HRP)-conjugated anti-dog IgG (Sigma Aldrich, USA) diluted 1:15,000 in PBST was added to each well and incubated at 37°C for 60 min. The content of the wells was discarded, and the plates were washed two times with PBST. Then, freshly prepared solution containing o-phenylene diamine (OPD) and 3% $H_2O_2$ (4 μL/mL of OPD) was added. The plate was incubated at room temperature for 15 min followed by addition of 2.5 N HCl to stop the reaction. Absorbance values were recorded at 492 nm wavelength using an ELISA reader. Percentage Positivity (PP) was calculated by the formula [61].

$$PP = \frac{\text{(OD of the test sample)}}{\text{(OD of positive control)}} X\,100$$

The positive cut-off PP value was determined by comparing test sera against different dilutions of pooled known vaccinated dog serum (control) having a titer of 8.0 IU/mL in each ELISA plate. The process was repeated 13 times on different days. The cut-off PP value was then derived by analyzing results of iELISA against RFFIT using Michaelis-Menton (2 parms) regression model which uses the general equation. Statistical analysis was performed by using R Studio 2022.12.0 Build 353 with R base version 4.2.2 using packages dplyr 1.0.10, ggplot2 3.4.0 and drc 3.0–1

$$y = \frac{ax}{b + x}$$

## Statistical analyses

The Graphpad prism software version 5.0 was used to calculate the mean, standard deviation, and standard error, as well as for curve fitting using non-linear polynomial quadratic equation. Spearman rank correlation was employed to compare the RFFIT and the ELISA. One-way ANOVA was employed to discern the effect of breed, gender, vaccine brand and association between time intervals of blood sampling and antibody titer, using GraphPad Prism V5 software. A p value of < 0.05 was considered statistically significant. Sensitivity and specificity of the ELISA was calculated as described previously [63], and by using the formulae as follows:

$$\text{Sensitivity} = \frac{a}{a + c} \; X \; 100$$

$$\text{Specificity} = \frac{d}{b + d} \; X \; 100$$

where
a = Number of samples positive by both standard test and test to be compared
b = Number of samples positive by test to be compared but negative by the standard test
c = Number of samples negative by test to be compared but positive by the standard test
d = Number of samples negative by both the tests.

Kappa value was also calculated as described previously [64].

# Results

The RABV *G* gene was amplified by PCR, and the resultant product was used in a second PCR to include an in-frame hexa-histidine (His) tag at the C terminus of the G protein. The product was inserted into pFastBac, and the cloning was confirmed by PCR. The construct pFastBac-RABV-G was used to transform DH10Bac cells, and the transposition of the *G* gene into the baculovirus genome was confirmed by blue-white colony screening, followed by PCR, and nucleotide sequencing.

Three different clones of pFastBac-RABV-G produced baculoviruses upon transfection of the Sf21 cells. When extracts and culture supernatants from uninfected cells and cells infected with one of the clones were subjected to PCR, both the pellet and the supernatant fractions from infected cells produced a band corresponding to the size (~ 1.6 kbp) of RABV *G* gene. As expected, the band intensity was higher with the infected cell pellet than the supernatant (Fig 1A; compare third and fourth lanes), reflecting the fact that much more DNA was present during virus replication in cells than that contained in the encapsidated virus released into the medium. On the other hand, only the pellet fraction of infected cells produced a band (Fig 1A, last two lanes), confirming that the mRNA would be present only in the fraction from cells where the virus would be expressing its genes, and not in the supernatant which is expected to contain mostly just the baculovirus. All the three different clones were then used to infect Sf21 cells, and the cell extracts were subjected to SDS-PAGE. A band of ~60 kDa was observed with all the three clones (Fig 1B, second to fourth lanes). When one of the clones was subjected to Western blotting with anti-His antibodies, a single band of ~60 kDa was observed with cells infected with the baculovirus encoding the *G* gene, and not with a control baculovirus which did not carry any insert (Fig 1C, compare the second lane against the first lane). These data together confirm the generation of the recombinant baculovirus and the expression of RABV G in Sf21 insect cells.

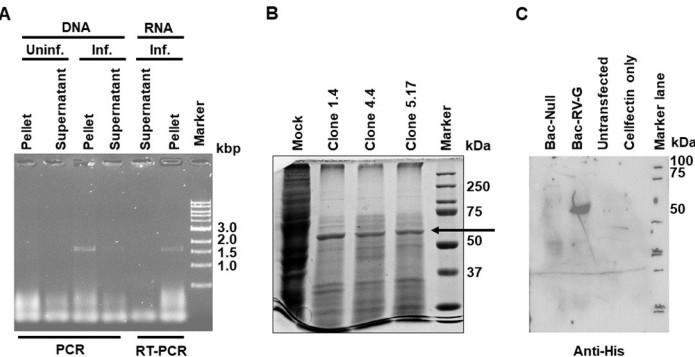

**Fig 1. Expression of G by recombinant baculovirus. A.** Uninfected (Uninf.) or infected (Inf.) Sf-21 cell cultures were separated into cells and supernatants and subjected to PCR or RT-PCR using primers for full-length *G* gene. **B.** Cell extracts of uninfected (Mock) and infected (three different clones designated 1.4, 4.4 and 5.17) were subjected to SDS-PAGE and Coomassie blue staining. The position of protein markers is shown on the right. **C.** Extracts from control (untransfected) and transfection control (Cellfectin only) as well as cells infected with a control baculovirus (Bac-Null) or a baculovirus encoding RABV *G* (Bac-RABV-G) were subjected to immunoblotting using anti-His antibodies. The position of protein markers is shown on the right.

Efforts were then made to purify G. Various lysis buffers and other conditions were initially attempted, and the procedure described in the Materials and Methods was finalized. Cell extracts were subjected to Ni-NTA chromatography, and a single band of expected size was observed in the elution fractions by SDS-PAGE followed by Coomassie blue staining (Fig 2A), as well as in pooled fractions by Western blotting using anti-His antibodies (Fig 2B). Furthermore, when pooled fractions were tested against sera from vaccinated and unvaccinated dogs, faint bands of expected size were observed only with the immune sera, and not with the non-immune sera (Fig 2C).

However, whilst G was consistently being expressed through the BVES, as evidenced through Western blotting of cell extracts using anti-His antibodies, the success rate of obtaining purified protein was poor. We frequently failed to observe any binding of the His-tagged protein to the Ni-NTA resin. Even when bound, often the yield of the purified protein was low, and insufficient to develop an ELISA. This was observed at two of the collaborating laboratories. Considering this, infected cell extracts were directly tested against some canine sera in

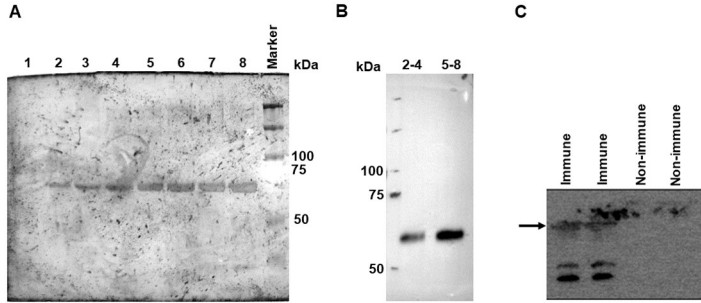

**Fig 2. Purification of G expressed through baculovirus. A.** Extracts of Sf21 cells infected with baculovirus encoding the RABV *G* gene were passed over a Ni-NTA column, and fractions (numbered 1 through 8) were subjected to SDS-PAGE and Coomassie blue staining. The position of protein markers is shown on the right. **B.** Pooled fractions (2–4 and 5–8) were subjected to immunoblotting using anti-His antibodies. **C.** The pooled fraction 5–8 was subjected to immunoblotting using sera from dogs which were either naïve (non-immune) or vaccinated (immune) against rabies. The expected position of the G protein is shown on the left.

an ELISA using anti-His antibodies. Following encouraging results, such extracts were used to develop the in-house iELISA. Total protein in the infected cell extract was estimated and following checker-board analysis, a concentration of 500 ng per well, test serum dilution of 1:100, and anti-dog-HRP conjugate of 1:15,000 were optimized. All the test serum samples were then subjected to iELISA as per the optimized protocol.

As evaluated against the reference standard serum by RFFIT, 210 out of the 245 serum samples (85.71%) showed an antibody titre of $\geq$ 0.5 IU/mL whereas the rest (n = 35; 14.29%) had a titre of < 0.5 IU/mL (S1 Table). When all the samples were tested in duplicate by iELISA, the PP values showed a certain range for each of the group of samples with a certain unitage. The number of samples falling under each of the titres by RFFIT as well as the corresponding range of PP values by iELISA are shown in Table 2.

The average PP values plotted against corresponding units of antibody titres generated a perfect hyperbola (Fig 3A), indicating concordance between the two tests. Spearman rank correlation analysis for all the 245 samples (Fig 3B) showed a coefficient (*r*) of 0.44 (P<0.0001) indicating a significant correlation between the two tests. The regression model fitted the results significantly with a = 114.572435 (Standard Error 1.910154), b = 1.255603 (Standard Error 0.062057) with very significant *p* value $\leq$ 2.2 x $10^{-16}$ (p < 0.0001). From the regression analysis, the positive cut-off value was found to be 32.64 (IU/mL) for the iELISA (equivalent to 0.5 IU /mL by RFFIT). The derived regression model was also used to convert the test results of iELISA values into predicted RFFIT values. The predicted and actual RFFIT values were plotted into Area Under the Curve (AUC) plot, which is shown in Fig 4.

When compared to RFFIT, the sensitivity and specificity of the ELISA were 90.48% and 80.00%, respectively (Table 3). Furthermore, the results of our in-house ELISA showed good agreement (κ value 0.647) with RFFIT (Table 3).

## Discussion

Detection and quantitation of antibodies in the sera of vaccinated humans and animals is essential for the assessing the success of vaccination and in turn the control of rabies. Owing to several constraints in performing RFFIT and FAVN for this purpose, ELISAs have been promulgated as alternatives. However, the development and application of a universally acceptable ELISA has been difficult. One of the impediments is the requirement for an appropriate antigen. Besides biosafety concerns for its production when requiring purification from whole virus, the G antigen may produce false positive results due to the presence of other viral antigens which may cross-react with antibodies to closely related viruses (unpublished observations). Hence, production of recombinant G is a safe and better alternative. We used baculovirus-infected insect cell extracts as antigen in an iELISA. The results showed good correlation with RFFIT.

**Table 2. Distribution of samples and their respective PP values under each of the titres.**

| Unitage (IU/mL) | No. of samples by RFFIT | Range of PP values by ELISA |
|---|---|---|
| 16.0 | 2 | 48.64–116.30 |
| 8.0 | 8 | 48.64–100.00 |
| 4.0 | 40 | 28.70–92.65 |
| 2.0 | 52 | 15.74–98.09 |
| 1.0 | 51 | 25.78–85.95 |
| 0.5 | 57 | 20.87–93.24 |
| <0.5 | 35 | 11.06–60.76 |

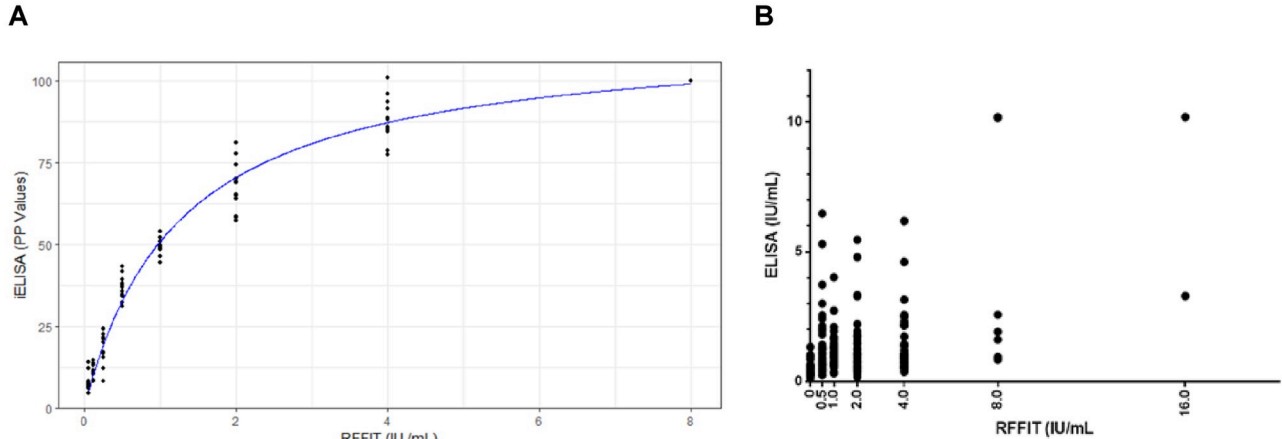

**Fig 3. Non-linear regression analysis of RFFIT vs iELISA. A.** Dog sera with an antibody titre of 8.0 IU/mL as estimated by RFFIT were pooled and serially diluted, and plotted against positive predictive values obtained by iELISA for the test samples. **B.** Values in IU/mL for the iELISA, derived based on the non-linear regression analysis from Fig 3, were plotted against values in IU/mL of RFFIT.

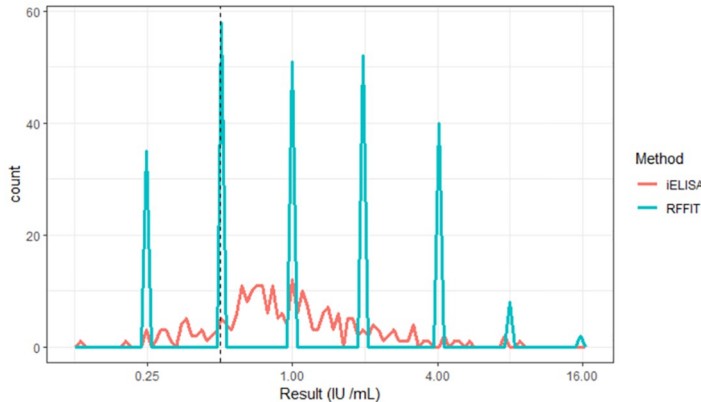

**Fig 4. Area under the curve for RFFIT vs iELISA.** The distribution of the frequency of the number of samples at each of the dilutions is plotted for RFFIT and iELISA. The dashed vertical line represents cut-off between positive and negative results at 0.5 IU/mL for RFFIT.

For any diagnostic test, one requires a continuous supply of sufficient quantities of an immunologically functional, native protein. However, the expression of viral glycoproteins is challenging due to their glycosylation and structural complexity. The RABV G is a type I membrane protein that assembles into trimers [65, 66]. Appropriate glycosylation and assembly of

**Table 3. Sensitivity and specificity of the in-house iELISA.**

| Test | | RFFIT | | Total | Sensitivity (%) | Specificity (%) | Kappa value |
|---|---|---|---|---|---|---|---|
| | | Positive | Negative | | | | |
| In-house ELISA | Positive | 190 | 7 | 197 | 90.48 | 80.00 | 0.647 |
| | Negative | 20 | 28 | 48 | | | |
| Total | | 210 | 35 | 245 | | | |

Note: Kappa (κ) value of 0.61–0.80 is considered as a substantial agreement between the two tests [29].

G is important for its proper expression, folding, oligomerization, and function [67–72]. The membrane anchorage of G on the virus surface and its trimeric structure are critical for the induction of NAbs [73–78]. The purification of native G in sufficient quantities depends on a number of variables such as the cell type, the extraction volume, the buffer used and its ionic strength, the concentration of salts, the detergent used and its concentration, the incubation temperatures and duration, and the propensity of the protein to refold [55, 73, 79–82]. Affinity tags (e.g. 6X His) fused to G have been reported but have produced variable recovery [55, 78, 80, 81], and may require additional or alternative methods such as gel filtration, ion exchange or lectin chromatography [35, 79, 82]. Indeed, we found it problematic to consistently purify His-tagged G from baculovirus-infected cell extracts. Other than the often-observed variations in the expression levels as well as the suitability of a lysis buffer to extract the protein, the major difficulty was the frequent inability to capture the protein on a nickel column even when expressed well. These were the reasons for us to use baculovirus-infected Sf21 cell extracts as antigen in the iELISA reported here.

The ELISA is a simple, rapid, affordable, and automatable test. The non-requirement for live virus makes ELISA a complementary or practical alternative test to neutralization methods for rabies. Several studies have reported the development of ELISAs for detecting antibodies to RABV, in human and animal sera [15, 19, 26, 28, 32–35, 37, 38, 40, 49, 83–86]. Indirect ELISA (iELISA) is the easiest of the ELISA methods to perform and standardize. However, its concordance with neutralization assays for anti-RABV antibodies has ranged from poor to excellent [22, 24, 26, 28–30, 34, 35, 40, 41, 45, 83, 86–91]. The variable performance of ELISAs could be due to the type of the antigen or the conjugated antibodies used, the species from which the serum is tested, the gold standard (MNT, RFFIT, FVNA or other neutralization tests) used for comparison, as well as inter-laboratory variations. Inconsistent relationship between neutralization (RFFIT) and antigen-binding (iELISA) either in the degree of agreement or in the direction of response has been noted for individual subjects [18, 19, 92]. Despite the limitations, the iELISA is the most amenable assay for the detection and quantitation of antibodies against RABV. A commercial iELISA is available, and has been tested with human as well as animal sera [33, 34, 49, 86, 93], although some results which are discrepant on sensitivity, specificity, accuracy, and seroconversion threshold in dogs vaccinated in the field as well as incompatible with clinical utility index have been noted [34, 47, 49, 94].

Rabies has the highest case fatality rate of any infectious disease. Hence, any assay which determines a cut-off value as a correlate of protection needs to be robust, particularly in eliminating false positive results. A serum antibody titre cut-off of 0.5 IU/mL is recognized as proof of adequate response deemed to be protective against rabies in cats, dogs, and wild carnivores [4, 16, 17, 19, 95]. However, this was originally determined based on neutralization methods such as MNT and RFFIT, for testing sera collected around one month after vaccination. To extrapolate this value to other methods requires extensive investigation and standardization, while also keeping in mind that the methods measure different kinds and characteristics of antibodies [18, 19]. In addition, the relationship between the measurement of binding antibody and neutralizing function is expected to vary between individuals and between different tests, and the comparisons may not be equal or sometimes even interpretable [18, 92]. The interpretation of the result therefore depends on the method applied, what it measures, the cut-off value and equivalency determination for the purpose of testing, and various sample considerations including quality, time of collection and the presence of toxic substances [18, 92].

Finally, several factors influence the development of any robust assay. These include (a) the fit for purpose, required sensitivity, specificity, limit of detection, limit of quantitation in the linear range, (b) the design of the experiment, (c) the type and the purity of the antigen, (c) the

type, source and quality of the test sample, (d) the type of capture and/or detection antibody (species, poly/monoclonal), (d) the conjugate and the detection system [94]. In terms of purity of the antigen used for coating, it has been observed that homogenous (purified) antigens are better for evaluating NAbs [96]. One of the major issues with ELISA is its reliability to faithfully discriminate sera with sufficient versus insufficient neutralization capacities. The fact that only a fraction of the anti-G MAbs can neutralize RABV suggests that conventional ELISAs may be inadequate to predict the quality or quantity of NAbs. Hence, even though several ELISAs have been developed, none of them have replaced the VNTs [97]. Therefore, ELISAs could be used as initial screening tests before applying the recommended tests for confirmation [98].

Our iELISA showed a sensitivity of 90.48% and a specificity of 80.00% against RFFIT, with a κ value of 0.647 with a sample size of 245 (dog sera). A few other studies have evaluated the application of different forms of ELISA in the context of monitoring antibodies in dogs. The earliest study tested an iEILSA using virus-derived G as the antigen against sera from wild and captive unvaccinated and vaccinated foxes [29, 30]. High correlation was observed, with r = 0.91 in one study and 0.88 in another, and agreement of 93.0% and 89.9%, respectively, against FAVN [29, 30]. Parallel testing at five different laboratories found that there was substantial to almost perfect agreement among the results between the tests as well as between ELISA and FAVN [29]. The test was later adapted with inactivated virus as antigen for testing 2360 sera from dogs and cats in three independent studies, where a specificity of 96.9% and sensitivity of 86.4% was reported [28]. A recently reported iELISA used detergent lysed virus from concentrated culture supernatant as antigen, and the test performed equally or better than two commercial iELISAs [34]. A competitive ELISA using virus-infected mammalian cell lysate as antigen, and 4350 vaccinated and unvaccinated dog sera, showed a correlation of 96.2% against FAVN [99]. A commercial blocking ELISA was tested in another study against 233 cat sera (46 unvaccinated, 187 vaccinated) and 765 dog sera (269 naïve and 496 vaccinated). The test showed high specificity (>97%) against a commercial iELISA, and 86.2% overall agreement with FAVN [22]. The sensitivity for various groups of sera, as categorized by different ranges of IU/mL, varied from 50% to 94.42%, and most of the discrepancy was with sera nearer to the cut-off value of 0.5 IU/mL [22]. The versatility of the blocking ELISA was demonstrated by applying a single test for sera from dogs, cats, cattle and humans, although the total number of samples (n = 46) was low; this test uses virus-like particles (VLP) obtained from mammalian cells [100]. Contrary to other studies where high specificity and lower sensitivity was observed, our results showed higher sensitivity and lower specificity. The reasons for this are unknown. It may be noted that all the above studies used different antigens (G purified from virus, inactivated virus, lysate of mammalian cells stably expressing G, lysate of infected mammalian cell culture supernatant, VLP), in different formats (iELISA, bELISA, cELISA) against samples from different locations from different species, whereas we used baculovirus-infected insect cell lysate as antigen against dog sera. In addition, the development stages vary from a proof-of-concept to interlaboratory validation. Hence, it is difficult to infer on the performance of our assay, which is still a proof-of-concept, in comparison to others.

Our results suggest that an iELISA using partially purified extracts of cells infected with baculovirus expressing RABV G could be of immense benefit to LMICs as it can be used in various laboratories, not only for the estimation of antibodies of pets intended for international travel but also for monitoring the success of mass immunization programmes. However, the requirement for infection of insect cells with baculovirus, the variable expression levels and consistent outcomes of purification remain as obstacles. Optimization of clones for consistent expression as well as alternative purification strategies may be useful in further improving of the assay.

## Supporting information

**S1 Table. Details of the samples with RFFIT and ELISA values.**
(DOCX)

**S1 Raw images.**
(PPTX)

## Acknowledgments

PJ and PG acknowledge fellowships provided respectively by the Council of Scientific & Industrial Research (CSIR) and the Department of Biotechnology (DBT), both under the Ministry of Science and Technology, Government of India.

## Author Contributions

**Conceptualization:** Madhusudan Hosamani, V. Balamurugan, R. Sharada, D. Rathnamma, Nagendra R. Hegde, Shrikrishna Isloor.

**Data curation:** A. K. Santosh, Deepak Kumar, Charanpreet Kaur, Priya Gupta, Pagala Jasmeen, L. Dilip, G. Kavitha, Suresh Basagoudanavar, Madhusudan Hosamani, R. Sharada, K. M. Sunil, Nagendra R. Hegde, Shrikrishna Isloor.

**Formal analysis:** A. K. Santosh, Suresh Basagoudanavar, Madhusudan Hosamani, R. Sharada, D. Rathnamma, Nagendra R. Hegde, Shrikrishna Isloor.

**Funding acquisition:** Shrikrishna Isloor.

**Investigation:** A. K. Santosh, Deepak Kumar, Charanpreet Kaur, Priya Gupta, Pagala Jasmeen, L. Dilip, G. Kavitha, Suresh Basagoudanavar, V. Balamurugan, R. Sharada, K. M. Sunil, Shrikrishna Isloor.

**Methodology:** A. K. Santosh, Deepak Kumar, Charanpreet Kaur, Priya Gupta, Pagala Jasmeen, L. Dilip, G. Kavitha, Suresh Basagoudanavar, V. Balamurugan, R. Sharada, K. M. Sunil, Shrikrishna Isloor.

**Project administration:** D. Rathnamma, Nagendra R. Hegde, Shrikrishna Isloor.

**Resources:** Deepak Kumar, Suresh Basagoudanavar, Madhusudan Hosamani, V. Balamurugan, R. Sharada, Nagendra R. Hegde, Shrikrishna Isloor.

**Supervision:** D. Rathnamma, Nagendra R. Hegde, Shrikrishna Isloor.

**Validation:** Suresh Basagoudanavar, Madhusudan Hosamani, R. Sharada, Shrikrishna Isloor.

**Writing – original draft:** A. K. Santosh, Nagendra R. Hegde, Shrikrishna Isloor.

**Writing – review & editing:** Nagendra R. Hegde, Shrikrishna Isloor.

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
