## [Decision Letter · Decision Letter 0]

7 Jun 2024

PONE-D-24-15752Evaluation of the immune status of dogs vaccinated against rabies by an enzyme-linked immunosorbent assay using crude preparations of insect cells infected with a recombinant baculovirus encoding the rabies virus glycoprotein genePLOS ONE

Dear Dr. Isloor,

Thank you for submitting your manuscript to PLOS ONE. After careful consideration, we feel that it has merit but does not fully meet PLOS ONE’s publication criteria as it currently stands. Therefore, we invite you to submit a revised version of the manuscript that addresses the points raised during the review process. As you can see the comments from our reviewers, they pointed out several major and minor issues for your manuscript. The editor also suggests that the figures (e.g., Fig.1 and Fig.2) need significant improvement to meet the standards and quality requirements of this journal. Please provide also the raw gel images and data points used for all figures. 

We look forward to receiving your revised manuscript.

Kind regards,

Jian Xu, Ph.D.

Academic Editor

PLOS ONE

2. In your Methods section, please provide additional details regarding participant consent from the owners of the animals. In the ethics statement in the Methods and online submission information, please ensure that you have specified (1) whether consent was informed and (2) what type you obtained (for instance, written or verbal). If the need for consent was waived by the ethics committee, please include this information.

Reviewers' comments:

Reviewer's Responses to Questions

**Comments to the Author**

1. Is the manuscript technically sound, and do the data support the conclusions?

Reviewer #1: Yes

2. Has the statistical analysis been performed appropriately and rigorously? 

Reviewer #1: Yes

3. Have the authors made all data underlying the findings in their manuscript fully available?

Reviewer #1: Yes

4. Is the manuscript presented in an intelligible fashion and written in standard English?

Reviewer #1: Yes

5. Review Comments to the Author

Reviewer #1: The article by Santosh et al., entitled "Evaluation of the immune status of dogs vaccinated against rabies by an enzyme-linked immunosorbent assay using crude preparations of insect cells infected with a recombinant baculovirus encoding the rabies virus glycoprotein gene," addresses the challenge of evaluating rabies vaccination effectiveness in dogs by proposing an ELISA as a practical alternative to the currently recommended tests, RFFIT and FAVN, which involve the handling of live rabies virus. The authors developed an ELISA using partially purified extracts of insect cells infected with baculovirus expressing the rabies virus glycoprotein (RV G). This method was evaluated against sera from vaccinated dogs, showing good concordance with RFFIT, with sensitivity and specificity of 90.48% and 80.00%, respectively. The proposed system offers a quick screening tool to determine the presence and approximate level of antibodies, and it holds potential for monitoring mass vaccination programs and certifying dogs for international travel. The study is well-conceived and contributes valuable insights into rabies vaccination monitoring. Thus, the manuscript can be accepted after minor revisions.

Minor Revisions

1. Introduction: Include the new species name Lyssavirus rabies according to ICTV (https://ictv.global/report/chapter/rhabdoviridae/rhabdoviridae/lyssavirus).

2. Sentence Improvement: Improve the sentence “However, there is no means to verify the success of the vaccination programmes other than keeping a record of the vaccination.” to provide a clearer context.

3. Cell Line Description: On page 10, line 206, specify which cell line was used and provide a detailed description.

4. Figure 1: Ensure the order of the description of the wells matches the order shown in the image to avoid confusion for the readers.

5. Figure 2C: Clarify which fraction was submitted by specifying “One of the fractions (indicate which) was submitted...”

6. Table 3: Include a footnote indicating that the kappa (κ) value of 0.61–0.80 is considered “substantial agreement” and not “almost perfect agreement.”

7. ELISA Antigen Explanation: Provide a better explanation as to why the purified G protein was not used to perform the ELISA, and describe the difficulties encountered in the purification step.

8. Comparison with Other Studies: Include a comparison with values found in other studies:

- Cliquet F, McElhinney LM, Servat A, Boucher JM, Lowings JP, Goddard T, Mansfield KL, Fooks AR (2004) Development of a qualitative indirect ELISA for the measurement of rabies virus-specific antibodies from vaccinated dogs and cats. J Virol Methods 117:1–8. https://doi.org/10.1016/j.jviromet.2003.12.001.

- Cliquet F, Müller T, Mutinelli F, Geronutti S, Brochier B, Selhorst T, Schereffer JL, Krafft N, Burow J, Schameitat A, Schlüter H, Aubert M (2003) Standardisation and establishment of a rabies ELISA test in European laboratories for assessing the efficacy of oral fox vaccination campaigns. Vaccine 21:2986–2993. https://doi.org/10.1016/S0264-410X(03)00102-6.

- Cliquet F, Sagné L, Schereffer JL, Aubert MF (2000) ELISA test for rabies antibody titration in orally vaccinated foxes sampled in the fields. Vaccine 18:3272–3279. https://doi.org/10.1016/S0264-410X(00)00127-4.

- Wasniewski M, Cliquet F. Evaluation of ELISA for detection of rabies antibodies in domestic carnivores. J Virol Methods. 2012 Jan;179(1):166-75. doi: 10.1016/j.jviromet.2011.10.019.

- Zajac MD (2019) Development and evaluation of a rabies enzyme-linked immunosorbent assay (ELISA) targeting IgM and IgG in human sera. https://doi.org/10.13140/RG.2.2.32365.54242.

- Zhao R, Yu P, Shan Y, Thirumeni N, Li M, Lv Y, Li J, Ren W, Huang L, Wei J, Sun Y, Zhu W, Sun L (2019) Rabies virus glycoprotein serology ELISA for measurement of neutralizing antibodies in sera of vaccinated human subjects. Vaccine 37:6060–6067. https://doi.org/10.1016/j.vaccine.2019.08.043.

9. Limitations of the Study: The authors should include a sentence in the discussion topic that reports the limitations of the study.

10. In the study by Kramps et al. (1994), the authors demonstrated that their blocking ELISA was superior to a commercially available indirect ELISA and to the 24-hour virus neutralization test in detecting low antibody levels in serum. Additionally, this blocking ELISA was capable of detecting specific antibodies in serum as early as 7 days post-infection. Thus, it would be beneficial to evaluate whether the newly developed ELISA here can detect antibodies as early as 7 days post-infection, similar to Kramps et al. studies.

Kramps JA, Magdalena J, Quak J, Weerdmeester K, Kaashoek MJ, Maris-Veldhuis MA, Rijsewijk FA, Keil G, van Oirschot JT. A simple, specific, and highly sensitive blocking enzyme-linked immunosorbent assay for detection of antibodies to bovine herpesvirus 1. J Clin Microbiol. 1994 Sep;32(9):2175-81. doi: 10.1128/jcm.32.9.2175-2181.1994.

Kramps JA, Banks M, Beer M, Kerkhofs P, Perrin M, Wellenberg GJ, Oirschot JT. Evaluation of tests for antibodies against bovine herpesvirus 1 performed in national reference laboratories in Europe. Vet Microbiol. 2004 Sep 8;102(3-4):169-81. doi: 10.1016/j.vetmic.2004.07.003.

6. PLOS authors have the option to publish the peer review history of their article (what does this mean?). If published, this will include your full peer review and any attached files.

Reviewer #1: No

---

## [Author Response · Author response to Decision Letter 0]

28 Oct 2024

Responses to reviewers

We thank the reviewer for a very detailed scrutiny of the manuscript. The suggestions have significantly improved the manuscript. Provided below are point by point responses to the suggestions.

Reviewer #1

1. Introduction: Include the new species name Lyssavirus rabies according to ICTV (https://ictv.global/report/chapter/rhabdoviridae/rhabdoviridae/lyssavirus). We have also changed the abbreviation of rabies virus from RV to RABV throughout the manuscript, in accordance with ICTV.

Response: This has been included right in the beginning of the Introduction (see line 47 in the clean copy of the revised manuscript).

2. Sentence Improvement: Improve the sentence “However, there is no means to verify the success of the vaccination programmes other than keeping a record of the vaccination.” to provide a clearer context.

Response: Context has been provided (see lines 55-58 in the clean copy of the revised manuscript)).

3. Cell Line Description: On page 10, line 206, specify which cell line was used and provide a detailed description.

Response: Cell line specification and description are now provided (see lines 247-250 in the clean copy of the revised manuscript)).

4. Figure 1: Ensure the order of the description of the wells matches the order shown in the image to avoid confusion for the readers.

Response: We thank the reviewer for pointing this out. The description has now been modified for all the three panels for clarity (see lines 298, 306 and 309 in the clean copy of the revised manuscript).

5. Figure 2C: Clarify which fraction was submitted by specifying “One of the fractions (indicate which) was submitted...”

Response: It was pooled fraction 5-8, and this has been specified now (see line 337 in the clean copy of the revised manuscript).

6. Table 3: Include a footnote indicating that the kappa (κ) value of 0.61–0.80 is considered “substantial agreement” and not “almost perfect agreement.”

Response: A foot note has been added to the Table (see lines 391-392 in the clean copy of the revised manuscript). It may be noted that we had stated it to be ‘good’ agreement in the description (see line 388 in the clean copy of the revised manuscript).

7. ELISA Antigen Explanation: Provide a better explanation as to why the purified G protein was not used to perform the ELISA, and describe the difficulties encountered in the purification step.

Response: A sentence is now included in the Discussion (see lines 419-423 in the clean copy of the revised manuscript).

8. Comparison with Other Studies: Include a comparison with values found in other studies:

- Cliquet F, McElhinney LM, Servat A, Boucher JM, Lowings JP, Goddard T, Mansfield KL, Fooks AR (2004) Development of a qualitative indirect ELISA for the measurement of rabies virus-specific antibodies from vaccinated dogs and cats. J Virol Methods 117:1–8. https://doi.org/10.1016/j.jviromet.2003.12.001.

- Cliquet F, Müller T, Mutinelli F, Geronutti S, Brochier B, Selhorst T, Schereffer JL, Krafft N, Burow J, Schameitat A, Schlüter H, Aubert M (2003) Standardisation and establishment of a rabies ELISA test in European laboratories for assessing the efficacy of oral fox vaccination campaigns. Vaccine 21:2986–2993. https://doi.org/10.1016/S0264-410X(03)00102-6.

- Cliquet F, Sagné L, Schereffer JL, Aubert MF (2000) ELISA test for rabies antibody titration in orally vaccinated foxes sampled in the fields. Vaccine 18:3272–3279. https://doi.org/10.1016/S0264-410X(00)00127-4.

- Wasniewski M, Cliquet F. Evaluation of ELISA for detection of rabies antibodies in domestic carnivores. J Virol Methods. 2012 Jan;179(1):166-75. doi: 10.1016/j.jviromet.2011.10.019.

- Zajac MD (2019) Development and evaluation of a rabies enzyme-linked immunosorbent assay (ELISA) targeting IgM and IgG in human sera. https://doi.org/10.13140/RG.2.2.32365.54242.

- Zhao R, Yu P, Shan Y, Thirumeni N, Li M, Lv Y, Li J, Ren W, Huang L, Wei J, Sun Y, Zhu W, Sun L (2019) Rabies virus glycoprotein serology ELISA for measurement of neutralizing antibodies in sera of vaccinated human subjects. Vaccine 37:6060–6067. https://doi.org/10.1016/j.vaccine.2019.08.043.

Response: We appreciate the reviewer for suggesting and drawing our attention to this. We have now added an entire paragraph in the Discussion on this (lines 471-503 in the clean copy of the revised manuscript), but have focussed on those studies which have tested for sera from animals, particularly carnivores. Although we agree with the importance of application of such tests in humans, in the context of our assay, studies using human sera only have been omitted.

9. Limitations of the Study: The authors should include a sentence in the discussion topic that reports the limitations of the study.

Response: Two sentences have been added at the end of Discussion (lines 508-511 in the clean copy of the revised manuscript).

10. In the study by Kramps et al. (1994), the authors demonstrated that their blocking ELISA was superior to a commercially available indirect ELISA and to the 24-hour virus neutralization test in detecting low antibody levels in serum. Additionally, this blocking ELISA was capable of detecting specific antibodies in serum as early as 7 days post-infection. Thus, it would be beneficial to evaluate whether the newly developed ELISA here can detect antibodies as early as 7 days post-infection, similar to Kramps et al. studies.

Kramps JA, Magdalena J, Quak J, Weerdmeester K, Kaashoek MJ, Maris-Veldhuis MA, Rijsewijk FA, Keil G, van Oirschot JT. A simple, specific, and highly sensitive blocking enzyme-linked immunosorbent assay for detection of antibodies to bovine herpesvirus 1. J Clin Microbiol. 1994 Sep;32(9):2175-81. doi: 10.1128/jcm.32.9.2175-2181.1994.

Kramps JA, Banks M, Beer M, Kerkhofs P, Perrin M, Wellenberg GJ, Oirschot JT. Evaluation of tests for antibodies against bovine herpesvirus 1 performed in national reference laboratories in Europe. Vet Microbiol. 2004 Sep 8;102(3-4):169-81. doi: 10.1016/j.vetmic.2004.07.003.

Response: Although we understand the argument for detection of antibodies as early as possible, we are not clear about the context of the studies suggested by the reviewer. They refer to bovine herpesvirus 1, and not to rabies. In addition, the context of the ELISA that we have developed is for monitoring antibodies after vaccination, whereas the mentioned studies about BHV-1 are after infection, which is impractical or inconsequential for rabies. The fundamental difference in these two cases is that the one for BHV-1 is for diagnostic purposes and the one for rabies is for monitoring purposes. Furthermore, early detection in case of rabies may not have a bearing on the potential to protect since early responses are dominated by IgM. It is thought that IgM is not as efficient as IgG against rabies virus due to its inability to penetrate interstitial tissue that has been impregnated with high levels of virus through the saliva in the bite of a rabid animal (Kennedy et al. 2007; Fooks 2018). Indeed, driving sustained IgG production is one of the main targets of vaccination against rabies (Flamand et al. 1993; Dorfmeier et al. 2013). Although we agree that a time-lapse experiment on the appearance of potentially protective antibodies is required, we feel that this discussion would be deviant from the focus of the manuscript.

References:

1. Dorfmeier, C.L.; Shen, S.; Tzvetkov, E.P.; McGettigan, J.P. 2013. Reinvestigating the role of IgM in rabies virus postexposure vaccination. J. Virol. 87:9217-9222.

2. Flamand, A.; Raux, H.; Gaudin, Y.; Ruigrok, R.W. 1993. Mechanisms of rabies virus neutralization. Virology 194:302-313.

3. Fooks, A. 2018. Rabies (Infection with RABV and Other Lyssaviruses). In Manual of Diagnotics Tests and Vaccines for Terrestrial Animals; World Organisation for Animal Health (OIE), Paris, France, 2018; pp. 578–612.

4. Kennedy, L.J.; Lunt, M.; Barnes, A.; McElhinney, L.; Fooks, A.R.; Baxter, D.N.; Ollier, W.E. 2007. Factors influencing the antibody response of dogs vaccinated against rabies. Vaccine 25:8500-8507.

Relevant Editorial Comments:

1. We note that you have included the phrase “data not shown” in your manuscript. Unfortunately, this does not meet our data sharing requirements. PLOS does not permit references to inaccessible data. We require that authors provide all relevant data within the paper, Supporting Information files, or in an acceptable, public repository. Please add a citation to support this phrase or upload the data that corresponds with these findings to a stable repository (such as Figshare or Dryad) and provide and URLs, DOIs, or accession numbers that may be used to access these data. Or, if the data are not a core part of the research being presented in your study, we ask that you remove the phrase that refers to these data.

Response: The phrase has been removed (see line 292 in the revised manuscript).

Response: Images of raw gels/blots are provided as a Supplementary file. It may be noted that panels 1B and 2A are already full images. The Supplementary file hence contains the images for panels 1A, 1B, 2B and 2C.

---

## [Decision Letter · Decision Letter 1]

12 Nov 2024

Evaluation of the immune status of dogs vaccinated against rabies by an enzyme-linked immunosorbent assay using crude preparations of insect cells infected with a recombinant baculovirus encoding the rabies virus glycoprotein gene

PONE-D-24-15752R1

Dear Dr. Isloor,

We’re pleased to inform you that your manuscript has been judged scientifically suitable for publication and will be formally accepted for publication once it meets all outstanding technical requirements.

Kind regards,

Jian Xu, Ph.D.

Academic Editor

PLOS ONE

Additional Editor Comments (optional):

Reviewers' comments:

Reviewer's Responses to Questions

**Comments to the Author**

1. If the authors have adequately addressed your comments raised in a previous round of review and you feel that this manuscript is now acceptable for publication, you may indicate that here to bypass the “Comments to the Author” section, enter your conflict of interest statement in the “Confidential to Editor” section, and submit your "Accept" recommendation.

Reviewer #1: All comments have been addressed

2. Is the manuscript technically sound, and do the data support the conclusions?

Reviewer #1: Yes

3. Has the statistical analysis been performed appropriately and rigorously? 

Reviewer #1: Yes

4. Have the authors made all data underlying the findings in their manuscript fully available?

Reviewer #1: Yes

5. Is the manuscript presented in an intelligible fashion and written in standard English?

Reviewer #1: Yes

6. Review Comments to the Author

Reviewer #1: The author has addressed all the questions and made the necessary revisions in the manuscript, which is now ready for acceptance in PLOS ONE.

7. PLOS authors have the option to publish the peer review history of their article (what does this mean?). If published, this will include your full peer review and any attached files.

Reviewer #1: No

---

## [Editor Report · Acceptance letter]

20 Nov 2024

PONE-D-24-15752R1 

PLOS ONE

Dear Dr. Isloor, 

I'm pleased to inform you that your manuscript has been deemed suitable for publication in PLOS ONE. Congratulations! Your manuscript is now being handed over to our production team.

Kind regards, 

on behalf of

Dr. Jian Xu 

Academic Editor

PLOS ONE